# Gestational toxoplasmosis treatment changes the child's prognosis: A cohort study in southern Brazil

**Ana Gabriela Gomes Ferrari Strang** [1,2]*, **Rafaela Gomes Ferrar**[3], **Ana Lúcia Falavigna-Guilherme**[4]

**1** Department of Medicine, Health Sciences Center, University Hospital of Maringá (HUM), State University of Maringá, Paraná (UEM), Brazil, **2** Postgraduate Program in Health Science, Health Sciences Center, State University of Maringá (UEM), Paraná, Brazil, **3** Postgraduate Program in Food Science, Institute of Chemistry, Federal University of Rio de Janeiro (UFRJ), Rio de Janeiro, Brazil, **4** Parasitology Laboratory, Department of Basic Health Sciences, State University of Maringá (UEM), Paraná, Brazil

* aggfstrang@uem.br

## Abstract

### Background

We evaluate the drug treatment for pregnant women with acute toxoplasmosis to reduce the risk of congenital infection, side effects (prenatal and postnatal treatment in children) and the hazard of discontinuing the infant's medication.

### Methods

We conducted a prospective cohort study to assess the risks of congenital toxoplasmosis among children born to acutely infected women with and without treatment. We examined the relationship between "exposed" and "infected children", "number of infant neutrophils", "prenatal" and "postnatal treatment". Factor analysis of mixed data (FAMD) was used to analyze the data. All children started treatment at the hospital.

### Findings

Between 2017 and 2021, 233 pregnant women were evaluated at the University Hospital of Maringá; ninety-four met criteria for acute gestational toxoplasmosis. We followed up 61 children; eleven (18%) had the infection confirmed and 50 (82%) were free of toxoplasmosis (exposed). Children born to untreated mothers have 6.5-times higher risk of being infected; the transmission rate among untreated mothers was 50% versus 8.3% among treated ones. Three decreasing values of immunoglobulin G were a security parameter for stopping the child's medication in the exposed group (50/61). Neutropenia was the leading side effect among children and the infected had a 2.7 times higher risk. There was no correlation between maternal use of pyrimethamine and children's neutropenia.

**Data Availability Statement:** All relevant data are within the paper and its Supporting Information files.

**Funding:** This work was supported by the State University of Maringá to AGGFS, RGF and ALFG. The funders had no role in study design, data collection and analysis, decision to publish, or preparation of the manuscript.

**Competing interests:** The authors have declared that no competing interests exist.

## Interpretation

The follow-up of women with acute *T. gondii* infection and their children, through a multidisciplinary team, availability of anti-*T. gondii* serology and pre- and post-natal treatments reduced the risk of toxoplasmosis transmission.

## Author summary

South America is the region with the highest burden of congenital toxoplasmosis and with the most pathogenic genotypes. Brazil is considered a country with a high prevalence of toxoplasmosis and a hotspot for outbreaks. The outcome of congenital transmission is influenced by factors such as *T. gondii* genotype, strain virulence, maternal immune status, maternal parasitemia, gestational age at the time of infection, and prenatal treatment. The last point mentioned is still controversial and raises several concerns, such as the financial impact of the monitoring and treatment of pregnant women and their children, the side effects of the medications used, and the benefit of investing in public health policies to mitigate toxoplasmosis. Information on the effectiveness of prenatal treatment is limited since it is challenging to carry out randomized studies. In the present manuscript, we evaluated the benefit of drug treatment for pregnant women with acute toxoplasmosis as a protective factor in reducing the risk of congenital infection. Also, we evaluated the side effects of prenatal and postnatal drug use in children and assessed the risk of discontinuing the medication in infants. We used the Strengthening the Reporting of Observational Studies in Epidemiology (STROBE) to report this study.

## 1. Introduction

Toxoplasmosis is a zoonosis caused by the intracellular protozoan *Toxoplasma gondii*, belonging to the phylum Apicomplexa. This pathogen is the most successful parasite; it has a worldwide distribution, can infect all warm-blooded animals and affects one-third of the human population [1].

The infection is often asymptomatic and the transmission is mainly foodborne and waterborne [2]. The primary vehicles are raw or undercooked meat with tissue cysts and water, raw vegetables and fruits with sporulated oocysts [3,4]. Additionally, in some cases of acute maternal infection, vertical transmission through the placenta to the fetus can be caused by *T. gondii* tachyzoites [5]. Congenital toxoplasmosis (CT) may lead to miscarriage, stillbirth, or sequelae in the child, such as retinochoroiditis and central nervous system lesions (i.e., cerebral calcification, hydrocephalus, mental retardation, epilepsy or psychiatric disease); however, the majority of congenitally-infected children appear asymptomatic at birth [6,7]. The outcome of congenital transmission is influenced by factors such as *T. gondii* genotype, strain virulence, maternal immune status, maternal parasitemia, gestational age at the time of infection and prenatal treatment. In Brazilian studies that identified the strain of *T. gondii* in children with congenital toxoplasmosis, it was possible to isolate fifteen different genotypes—#11 (BrII), #8 (BrIII), and the atypical #36, #41, #67, #108, #162, #166, #206, #207, #208, #209, #210, #211, #212 [6].#36, #41, #67, #108, #162, #166, #206, #207, #208, #209, #210, #211, #212 [6].

The risk of vertical transmission increases with the evolution of pregnancy: less than 10% before 12 weeks, 15–20% at 13–20 weeks, 44% at 26 weeks and 71% at 36 weeks [8,9].

### The size of the problem and the consequences of congenital CT

The estimated global incidence of CT is 190,100 annual cases; this corresponds to an incidence rate of approximately 1.5 cases in 1,000 live births [10]. In Brazil, the reported prevalence ranges from 0.1 cases to 3.4/1,000 live births and studies have estimated that approximately 35% of children had neurological diseases including hydrocephalus, microcephaly and mental retardation, 80% had ocular lesions and one report mentioned 40% of children with hearing loss [6,11].

### Strategies to mitigate the impact of gestational and congenital toxoplasmosis

Prenatal treatment for Toxoplasma infection aims to prevent neurological or visual impairment by the reduction of mother-to-child transmission of infection or, once the fetal infection has occurred, by limiting cell damage caused by the parasite [12,13]. Information on the effectiveness of prenatal treatment is limited since it is challenging to carry out randomized studies with untreated versus treated groups [12]. However, it is possible to evaluate cases in which pregnant women were not treated as they arrived late to reference centers, missed appointments, or did not perform adequate prenatal care. The prenatal approach relies on the hypotheses that maternal treatment reduces the risk of mother-to-child transmission and congenital infection treated prenatally is associated with a lower risk of severe lesions [14,15]. Neonatal diagnosis can be performed by collecting anti-*T. gondii* serology in the maternity ward or through universal blood screening on filter paper. This neonatal screening was implemented in Denmark and is carried out in the states of Massachusetts and New Hampshire in the United States and Colombia [16–19]. The neonatal test using dried blood on filter paper became mandatory throughout Brazil in 2021 [20].

### Brazil's strategies

In Brazil, due to the high prevalence of toxoplasmosis, screening begins in the prenatal period to reduce the lifetime consequences of CT [21,22]. Since 2015, there has been a working group in the Brazilian Ministry of Health for the construction of integrated surveillance on gestational, congenital and acquired toxoplasmosis [23]. In 2016, gestational and CT became a disease of compulsory notification, which means an advance in the control of this pathogen once we understand better the impact that this disease has on the population [23]. Many advances have been achieved. The main purpose of screening is to identify susceptible pregnant women (without previous infection) for monitoring during pregnancy. Early detection aims to prevent fetal transmission and provide necessary gestational and postnatal treatment. All susceptible pregnant women (not infected) must perform at least three serology tests during pregnancy (in the first, second and third gestational trimesters) and another serology is recommended at the time of delivery or during the puerperium (on the maternity ward). The confirmed cases are referred for high-risk prenatal care. Ideally, serology for toxoplasmosis must be known in women before conception. Until 2018, there were no standardized guidelines for carrying out Brazil's surveillance. However, some states of the union have developed protocols based on the local prevalence of the disease. The diagnosis is mainly based on indirect methods, such as serology (serology for the detection of IgG, IgM, IgA and the determination of IgG avidity anti-*T. gondii*), but also in methods of direct detection, such as molecular polymerase chain reaction techniques (real-time PCR and PCR), parasite isolation (cell cultures and inoculation in mice) and histological or immunohistological studies of the parasite [23]. The main methods used to confirm cases by the Brazilian Central Public Health Laboratories have been IgM,

IgG serology and IgG avidity (until 16 gestational weeks). Eventually, they perform PCR according to laboratory capacity. Sometimes, it is necessary to combine different methods to achieve the proper assessment [23]. In 2021, the Ministry of Health included anti-*T. gondii* IgM serology in the heel prick test, which provides a greater chance of diagnosis in children despite maternal tests, given that it is performed universally (all children born in hospitals must have the heel prick test before discharge) [20].

Despite the advances, many challenges still need to be overcome, such as the implementation of public health services in the countryside, adequate training of health professionals to interpret laboratory results, guaranteeing the supply of drugs against gestational and congenital toxoplasmosis and, above all, providing evidence for the effectiveness of prenatal or postnatal treatment of toxoplasmosis.

In this study, we evaluated the benefit of drug treatment for pregnant women with acute toxoplasmosis as a protective factor in reducing the risk of congenital infection. Also, we evaluated the side effects of prenatal and postnatal drug treatment in children and assessed the risk of discontinuing the drug treatment in infants after two decreasing IgG anti-*T. gondii* serology tests. Finally, we describe clinical and radiological findings in children with congenital toxoplasmosis.

## 2. Material and methods

### Ethics statement

The Ethics and Research Committee of the State University of Maringa approved this study (CAAE N° 09286918.5.0000.0104). The informed consent term was not obtained because the participants were kept anonymous.

### Study design

The study was conducted from April 1, 2017, to December 31, 2021. This cohort study included all children born to women with confirmed acute toxoplasmosis (positive IgM and IgG anti-*T. gondii*) during prenatal care or diagnosed during delivery at the University Hospital of Maringá (HUM) University State of Maringá (UEM), South of Brazil. The HUM is certificated as a "Baby-friendly Hospital." We used the Strengthening the Reporting of Observational Studies in Epidemiology (STROBE) to report this study.

### Study population

The University Hospital of Maringá is located in the Northwest region of the state of Paraná. It is a teaching hospital that belongs to the State University of Maringá and provides care to 30 municipalities, which covers a population of approximately 800,000 inhabitants. Since 2005, the Hospital has offered specialized care to pregnant women suspected of having acute gestational toxoplasmosis through a high-risk outpatient clinic. In 2017, the hospital opened an exclusive outpatient clinic named "ToxoPed" for the follow-up of children born to these mothers.

Between 2017 and December 2021, 233 pregnant women were evaluated at the University Hospital of Maringá for suspected acute *T. gondii* infection. After analyzing and checking the exams (anti-*T. gondii* serologies, gestational ultrasounds) collected during the current pregnancy and evaluating the previous history of these women, we concluded that 130 had chronic toxoplasmosis (acquired before the current pregnancy), nine did not have the disease (they were referred due to an error in the interpretation of the serological exams) and 94 met the criteria for acute gestational toxoplasmosis.

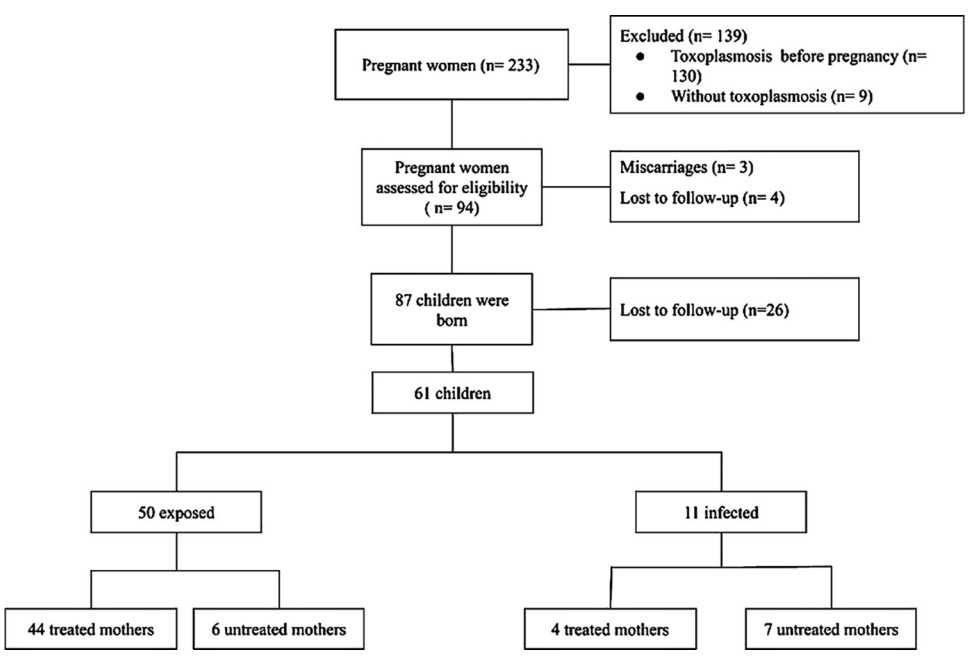

**Fig 1. Number of women and children evaluated for research.**

Among the 94 acutely infected pregnant women, four children were born in particular hospitals and the mothers did no follow-up in our pediatric service. Twenty-six children were lost to follow-up during the COVID-19 pandemic and their outcome was not concluded.

There were three spontaneous abortions, where it was not possible to assess the presence of the parasite by the pathological anatomy. The family of one of the mothers who had an abortion was found to be infected (her three children and her husband); our study group evaluated the case, but we could not prove the source of infection.

Finally, we had the opportunity to follow-up 61 children in the "ToxoPed" outpatient clinic to conclude whether there was transplacental transmission of the parasite (Fig 1).

Two groups of children were defined during follow-up. The first group was presumably infected children, called "exposed children". They presented negative serology for anti-*T. gondii* IgM, were IgG positive at birth and showed unaltered cerebrospinal fluid, physical examination, cranial computed tomography and retinal examinations. In this group, the IgG decreased and became negative during the first year of life and these children were considered free of toxoplasmosis. The second group was the children who were definitely infected during pregnancy. The following criteria were used to consider a child infected: i) positive anti-*T. gondii* IgM; ii) negative anti-*T. gondii* IgM but rising anti-*T. gondii* IgG; and iii) eye or brain lesions during follow-up.

Among the definitely infected children, we considered those who presented altered serology but unaltered brain tomography and no eye lesions to be asymptomatic.

Exclusion criteria were: children born to pregnant women with primary or acquired imunosupression, children with diseases or who used medications that could confuse the data analysis and children born to mothers who were chronically infected with *T. gondii*.

## Laboratory procedure, follow-up and definitions

In the HUM/UEM high-risk prenatal care outpatient clinic, all of the pregnant women who confirmed acute toxoplasmosis received spiramycin when the infection was diagnosed in the

first trimester and after 34 weeks of gestation, from 16 to 34 weeks of gestation, the women were treated with a combination of sulfadiazine, pyrimethamine and folinic acid. According to the institutional protocol (Table A in S1 File), all children started treatment at the maternity ward (rooming in) and were discharged using pyrimethamine, sulfadiazine and folinic acid. They had outpatient follow-ups every 45 days until they had completed 90 days. This way, the children had three serologies by the time they reached 90 days of life. The first serology was collected in the maternity ward after 24 hours of life (we did not use umbilical cord blood), the second when they were 30–35 days old and the third collected after 50 days. If the child presented a decrease in anti-*T. gondii* IgG serology at 90 days compared to the previous two readings, the medication was suspended and the follow-up continued every two months until one year. Children who were negative or had a zero value for anti-*T. gondii* IgG at 12 months old were defined as free of disease (the first group, "exposed"). Children considered infected received anti-Toxoplasma medication until 12 months of age. After reaching the age of one year, the medication was discontinued. All children can continue their follow-up at the "ToxoPed" outpatient clinic until they are 15 years-old. The intention is to follow these groups of children over time and study their neuropsychomotor development. Right now, our oldest patients are approximately 5 1/2 years old. Between 2017–2018, the amplified chemiluminescent method was used with Vitros ECiQ Toxoplasma IgG and IgM assays (Ortho-Clinical Diagnostics), according to the manufacturer's specifications. Since December 2019, we have used microparticle immunoassay with ARCHITECT Toxoplasma IgG and IgM kits, according to the manufacturer's specifications (Abbott, Wiesbaden, Germany).

In the group of infected children, after reaching the end of their first year of life, the medication was discontinued and the children continued with retinal assessments every six months in the first two years and were followed-up on psychomotor development. The social workers scheduled the child's appointments in cases of absence. Neutropenia was considered when the number of neutrophils fell below 1,000/mm3.

## Statistical analysis

The relative risk (RR) was calculated to assess whether prenatal treatment modifies the outcome in the child and to analyze the association between pediatric neutropenia and the condition "exposed" or "infected". The significance of Fisher's exact test level was $p < 0.05$ with a 95% confidence interval. Logistic regression was calculated to assess whether the use of pyrimethamine by the mother was associated with pediatric neutropenia (Tables B, C, and D in S1 File).

We examined the relationship between exposed children, infected children, the number of infants' neutrophils, maternal treatment and postnatal treatment. We used factor analysis of mixed data (FAMD) to analyze the data set. FAMD is a principal components method to describe, summarizes and visualize multidimensional matrix with mixed data. As any principal components method, its aim is to study the similarities between individuals, the relationships between variables and to link the study of the individuals with the variables. Such methods reduce the dimensionality of the data and provide subspace that best represent the data. Briefly, this statistical method makes possible to analyze the similarity between individuals by considering mixed types of variables. We explore the association between all variables (Tables E and F in S1 File). In the multifactorial evaluation, the quantitative variables were grouped into dimensions 1 and 2 according to their contribution value (Figs A and B in S1 File).

After analyzing the variables, we built a decision tree to verify the safety of suspending the medication after three decreasing values of IgG anti-*T. gondii*; for this, we used the Rpart library. All analyses used R 4.0.2 software (R Foundation for Statistical Computing, Vienna,

Austria). The principal libraries used were: factoextra, FactoMineR, ggplot2, dplyr and janitor. To correct the missing values, we used the algorithms contained in the input PCA of the R documentation [24,25]. The data is available on S1 Data.

## 3. Results

The follow-up was performed in 61 children to conclude whether there was transplacental transmission of the parasite: eleven (18%) had the infection confirmed (infected) and 50 (82%) were free of toxoplasmosis (exposed). Of the 50 who were disease-free, 44 were born to treated women and six were untreated (diagnosed during hospitalization for childbirth). Among the eleven infected children, seven were born to untreated women (Fig 1). Among the untreated women, only two did not have prenatal care and the others had negative third trimester tests. We found an RR = 0.15 among infected children which means that children born to untreated mothers have 6.5-times higher risk of being infected (p = 0.0006, CI 95% 0.05–0.44). The transmission rate among untreated mothers was 50% versus 8.3% among treated ones. All children started drug treatment in the maternity ward and 53/61 were suspended after three reducing serological tests (anti-*T. gondii* IgG). Among the eleven infected children, three were detected after discontinuing the medications; their anti-*T. gondii* IgG increased and the anti-*T. gondii* IgM remained negative; these three children remained asymptomatic. Thus, withdrawal of the medication did not cause clinical complications. The decision tree shows that three decreasing readings of anti-*T. gondii* IgG is a security parameter for stopping the child's medication in the exposed group (Fig D in S1 File). Those considered exposed (50/61) became negative for anti-*T. gondii* IgG at an average of 9 months (ranging from 3–12 months).

Ten infected children completed treatment for up to 12 months and one infected child failed to continue drug treatment due to serious neutropenia. During the children's follow-up, 59 children were allowed to analyze their blood counts serially and only neutropenia was significant among the other side effects (vomiting, nauseous, cutaneous rash, anemia, thrombocytopenia). We observed statistical significance in pediatric neutropenia (21/59; 35.6%) and the infected group (RR = 0.37 CI 95%: 0.20–0.67 p = 0.001). Children in the infected group are 2.7-times more likely to develop neutropenia.

All children who had neutropenia were under outpatient follow-up and did not have other diseases that could cause a low number of neutrophils. No neutropenic child was hospitalized when the blood test was altered. Among the infected patients who developed neutropenia, three were asymptomatic, and five were symptomatic (1 with ocular lesions and brain calcifications, 2 with brain calcifications, and 2 with ocular lesions). There was no correlation between maternal use of pyrimethamine and the presence of neutropenia in children (Table 1).

Asymptomatic infected children (5/11) only had altered plasma serology (positive anti- *T. gondii* IgM and IgG) without clinical or radiological manifestations (calcifications, retinochorioditis). Among the asymptomatics, three mothers were diagnosed in the maternity ward, so they did not receive any drug treatment. The other two mothers were diagnosed in the third trimester and started medication, four and seven weeks from diagnosis. Six children had symptoms, three with cerebral calcifications, one with retinochorioditis and two with both. Gestational toxoplasmosis was detected in all trimesters but especially in the third one (Table 1).

We could make a real distinction between the two groups after FAMD. In Fig 2A, we can observe the formation of two groups; in dimension 1 (Dim1), the variables "no maternal treatment" (Mno), "maternity", "Neutropenia yes" (Nyes) group the infected individuals. In dimension 2 (Dim2), there is a closeness between the variables "second trimester" (2t), "third trimester" (3t), "first trimester" (1t), and the exposed individuals. In Fig 2B, the exposed and

**Table 1. General characteristics and statistical analysis.**

| General characteristics | | | | N(%) |
|---|---|---|---|---|
| **Child** | | | | |
| male | | | | 31 (50.8%) |
| female | | | | 30 (49.2%) |
| weigth birth | 3,240* | (2,810–3,520)** | | 53 (86.9%) |
| head circunference at birth | 34 | (33–35) | | 45 (73.8%) |
| protein in CSF | 142 | (48–540) | | 44 (72,1%) |
| **Child neutropenia** | | | | |
| No | | | | 38 (62.3%) |
| Yes | | | | 21 (34.4%) |
| no avaliable | | | | 2 (3.8%) |
| **Mothers** | | | | |
| **Gestacional trimestre diagnosis** | | | | |
| 1st | | | | 14 (23%) |
| 2nd | | | | 23 (38%) |
| 3th | | | | 14 (23%) |
| Maternity ward | | | | 10 (16%) |
| **Mother drug scheme** | | | | |
| Protocol A | | | | 20(33%) |
| Protocol B | | | | 17(28%) |
| Protocol C | | | | 11(18%) |
| **Statical analisys** | | | | |
| **Infected children and** | | RR[1] | 95% CI[1] | p-value[1] |
| no mother treatment | | 0.15 | 0.05–0.45 | 0.0006 |
| neutropenia < 1000/mm3 | | 0.37 | 0.20–0.67 | 0.0010 |
| **Mother pyrimethamine use and** | | log(OR)[2] | 95% CI[1] | p-value[2] |
| children neutropenia < 1000/mm³ | | 0.6 0.4 | -0.71–2.0 | 0.4 |

[1] n (%), CI = Confidence Interval, CSF = cerebrospinal fluid

1t = first trimester, 2t = second trimester, 3t = thirst trimester

Protocol A = spiramycin

Protocol B = sulfadiazine, pyrimethamine, folinic acid + spiramycin

Protocol C = spiramycin + sulfadiazine, pyrimethamine, folinic acid + spiramycin

p-value[1] = P value were calculated by Fisher- test

[2] OR = Odds Ratio

p-value[2] = P value were calculated by Logistic regression

*mean

** max-min

weigth in gramas, head circunference in centimeters, protein in miligramas/deciliters

infected groups are well characterized visually. In this figure, each individual is represented by a dot. In Fig 2C, we visualize the individuals grouped by the variables that describe them.

## 4. Discussion

This study evaluated prenatal care and follow-up of children in southern Brazil. The results reinforce the importance of treating acute toxoplasmosis in the gestational period, with statistical evidence and more risk between the infected group and the absence of maternal treatment. Our data corroborate those of studies in other regions confirming the importance of toxoplasmosis screening and treatment during prenatal care and at the time of delivery to achieve the

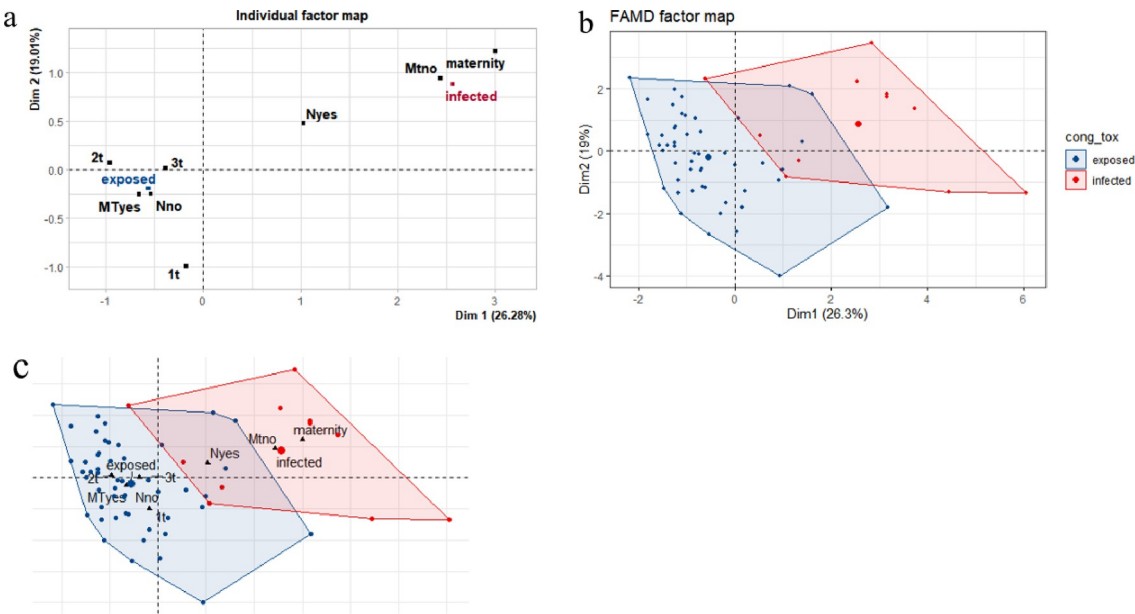

**Fig 2. Correlation among quantitative and qualitative variables.** 2a: the variables with the highest correlation are closer to each other, we can observe a group (Dim 1) formed by the infected children (infected), untreated mothers (Mtno) and the diagnosis performed at the hospital maternity (maternity). The other group (Dim 2) includes children who have not confirmed congenital infection (exposed), maternal treatment (MTyes), children without neutropenia (Nno) and diagnoses performed in the second, third and first trimesters (1t, 2t, 3t). In Fig 2B, the individuals are represented by dots. In 2c individuals dots) are grouped by the variables.

best outcomes for pregnant women and their babies [26–29]. It is necessary to emphasize that the speed when starting treatment (pre- or post-natal) increases the chances of acting on tachyzoites [8,29]. As we know, spiramycin, pyrimethamine and sulfadiazine, the current standard therapy for toxoplasmosis, can suppress tachyzoite growth (the acute life cycle stage) but do not affect bradyzoites [30,31]. Tachyzoites are responsible for congenital infection when they are able to overcome the placental barrier. In this way, women infected during pregnancy (or around conception) are generally offered spiramycin, a potent macrolide antibiotic that concentrates in the placenta, making it an ideal preliminary treatment option for maternal-fetal transmission prevention. Unfortunately, spiramycin is ineffective for treating established fetal infection since it uncrosses the placental barrier [32]. Once the fetal infection is established, it is necessary to use medications to reach the fetus. In this context, the combination of sulfadiazine, pyrimethamine and folinic acid (SPF) is the recommended therapy [30,33,34].

A critical aspect observed in our research was that there was a higher risk of neutropenia in infected infants. It is important to emphasize that the comparison between neutrophil blood counts of both groups (exposed and infected) was made with samples collected in the first three months of life. In other words, all babies were under an anti-parasitic regimen. Concerning this, it is known that sulfadiazine and pyrimethamine are capable of causing neutropenia by acting in a deleterious manner on the bone marrow by the action of the drug on the folate cycle during DNA division. Therefore, we did not find any association between maternal use of SPF and neutropenia in the child, which makes us think about the impact of the parasite itself. An experimental study showed that mice infected by *T. gondii* (RH strain) presented leukocyte depletion in the bone marrow and peripheral blood [35]. They proposed that this neutropenia might be associated with the proliferative action of tachyzoites (cell lysis) and possible changes in other points of host cell metabolism. In this way, we postulated that the parasite, itself, might cause a higher risk of neutropenia in infected children. However, further human studies are needed to prove this point of view.

Among untreated women, only two did not have prenatal care and the others had negative third trimester tests. Nevertheless, when collected at the maternity hospital, they showed infection acquired in the last weeks of pregnancy. This finding corroborates the need for universal monthly testing of susceptible pregnant women and maternity wards, as in other countries such as France and Italy. The testing of mothers in hospital allows the detection of infected children who would otherwise be lost. The serology results (anti-*T. gondii* IgM and IgG) take 48 to 72 hours to be ready, at which point we promptly began the children's treatment. It is worth remembering that in Brazil, in 2021, there was a significant advance in the diagnosis of acute toxoplasmosis in children because the serological research on this zoonosis was included in the heel prick test. However, compared to the maternity test, the heel test can take up to 30 days to deliver results. Postnatal treatment has also decreased the development and worsening of new signs and symptoms. Bearing this and given the presence of more virulent strains in Brazil, that may cause tissue damage in infected children, our protocol is to discharge all children born to a mother with acute toxoplasmosis with triple treatment (SPF).

Ruling out pediatric infection is not easy because only a negative IgG test allows us to say that the child is free of disease [8]. The multidisciplinary monitoring associated with three sequential dosages of immunoglobulin G, with an interval of 30 days between them, proved to be a safe parameter for the suspension of the medication in exposed (uninfected) children. In addition to reducing adverse effects, it can save costs and reduce the burden of tasks that a mother has with her baby. We must consider that administering three drugs with different dosages more than once a day and every day of the week requires a considerable commitment for women who already have many tasks with their babies.

The limitations of this study were i) it was not possible to identify whether there was a difference in the prenatal transmission according to prophylactic treatment, spiramycin, or triple schedule; ii) the study has primarily captured data from children who attended public health services, meaning that children who were not seen in the public health services would have been undetected; iii) the COVID-19 pandemic caused losses and follow-up discontinuity, leading to data loss; and iv) despite being a representative sample for Southern Brazil, it is still a limited sample, requiring multicenter studies.

Through the multidisciplinary follow-up of pregnant women and their children, the use of serological and radiological tests, and the opportunity for drug treatment, a reduction in the risk of congenital toxoplasmosis transmission was verified. With careful follow-up, it was possible to suspend the drug in exposed children and maintain treatment only in infected children, reducing side effects and costs. Future studies must better elucidate neutropenia among infected children and clarify the relationship between neutrophils and *Toxoplasma gondii*.

## Supporting information

**S1 File. Care protocol, tables and images of statistical analysis.**
(DOCX)

**S1 Data. Data obtained from monitoring patients.**
(XLSX)

## Author Contributions

**Conceptualization:** Ana Gabriela Gomes Ferrari Strang, Rafaela Gomes Ferrar, Ana Lúcia Falavigna-Guilherme.

**Data curation:** Ana Gabriela Gomes Ferrari Strang.

**Formal analysis:** Ana Gabriela Gomes Ferrari Strang.

**Investigation:** Ana Gabriela Gomes Ferrari Strang.

**Methodology:** Ana Gabriela Gomes Ferrari Strang.

**Project administration:** Ana Gabriela Gomes Ferrari Strang.

**Resources:** Ana Gabriela Gomes Ferrari Strang, Ana Lúcia Falavigna-Guilherme.

**Supervision:** Rafaela Gomes Ferrar, Ana Lúcia Falavigna-Guilherme.

**Validation:** Rafaela Gomes Ferrar, Ana Lúcia Falavigna-Guilherme.

**Visualization:** Ana Gabriela Gomes Ferrari Strang, Rafaela Gomes Ferrar, Ana Lúcia Falavigna-Guilherme.

**Writing – original draft:** Ana Gabriela Gomes Ferrari Strang, Rafaela Gomes Ferrar, Ana Lúcia Falavigna-Guilherme.

**Writing – review & editing:** Ana Gabriela Gomes Ferrari Strang, Rafaela Gomes Ferrar, Ana Lúcia Falavigna-Guilherme.

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
