## [Decision Letter · Decision Letter 0]

23 Jan 2023

Dear Mrs strang,

Thank you very much for submitting your manuscript "Gestational toxoplasmosis treatment changes the child's prognosis: a cohort study in southern Brazil" for consideration at PLOS Neglected Tropical Diseases. As with all papers reviewed by the journal, your manuscript was reviewed by members of the editorial board and by several independent reviewers. In light of the reviews (below this email), we would like to invite the resubmission of a significantly-revised version that takes into account the reviewers' comments. 

We cannot make any decision about publication until we have seen the revised manuscript and your response to the reviewers' comments. Your revised manuscript is also likely to be sent to reviewers for further evaluation.

Sincerely,

Hamed Kalani

Academic Editor

Ricardo Fujiwara

Section Editor

Reviewers' comments:

Reviewer 1:

1. In the field of sample size and compliance with ethical requirements, there is a little ambiguity, which should be given more comprehensive explanations and evidence by the authors. 

2. Toxoplasma gondii should be written in italics throughout the manuscript.

Reviewer 2:

1. The authors need to include the information on how the serological tests to detect IgG and IgM antibodies to T. gondii were performed? Did they use commercial kits for this purpose? If this is the case, they need to provide name of the kits and name of the companies that provide the kits.

2. The first paragraph of the Results section (Page 11): The authors state “All children started drug treatment in the maternity ward and 52/61 were suspended after three reducing serological tests (anti-T. gondii IgG)”. This statement makes readers to consider that 9 (61-52 =9) children showed serological evidence for having congenital infection. The authors also state “Among the eleven infected children, three were detected after discontinuing the medications”. These statements together make readers to consider 12 (9 + 3= 12) children were congenitally infected. However, the actual number of infected children was 11. The authors need to provide additional information to explain these numbers.

3. In relation to neutropenia in congenitally infected children, the authors referred an article an occurrence of neutropenia in a murine model of acute T. gondii infection. The authors need to include the information that this murine model is a lethal infection of animals with the virulent RH strain of the parasite. The authors also need to be more careful about applying the information from this murine model of the lethal infection to congenitally infected children.

4. Among 11 congenitally infected children, 5 were asymptomatic. It is important to provide the information on whether treatment of their mothers makes any association to determine their infected children become symptomatic or asymptomatic.

5. It is also important to perform the analysis on whether neutropenia is associated with symptomatic infection or not.

6. The authors need to provide more detailed descriptions in the description in the last paragraph of the “Results” section. This description needs to include what each of Figs. 2a, 2b, and 2c indicates and how these figures need to be read. It is also helpful to explain what each of Supplemental Figures 1, 2, and 3 indicates.

7.The end of the first paragraph on page 16: The meaning the following sentence is unclear; “Bearing this in mind and given the data that point to the presence of more virulent strains in Brazil (greater chance of tissue damage in infected children), the protocol is to discharge all children born to a mother with acute toxoplasmosis with triple treatment (SPF).”

8. In Supplemental Tables, all abbreviations (e.g. MTyes) need to be spelled out at the bottom of each Table.

9. It is unclear what Supplemental Figure 4 indicates.

10.. It will be helpful to include the information on what genotypes of T. gondii are abundant in the southern Brazil, where this clinical study was performed. 
---

## [Editor Report · Decision Letter 1]

23 May 2023

Dear Mrs strang,

Thank you very much for submitting your manuscript "Gestational toxoplasmosis treatment changes the child's prognosis: a cohort study in southern Brazil" for consideration at PLOS Neglected Tropical Diseases. As with all papers reviewed by the journal, your manuscript was reviewed by members of the editorial board and by several independent reviewers. The reviewers appreciated the attention to an important topic. Based on the reviews, we are likely to accept this manuscript for publication, providing that you modify the manuscript according to the review recommendations. 

The authors responded to the comments, and the manuscript needs to be carefully checked in terms of writing (for example: line 219 has a dot added after the parentheses and should be deleted) and the sentences need to be strengthened in terms of academic writing.

Sincerely,

Hamed Kalani

Academic Editor

Ricardo Fujiwara

Section Editor

The authors responded to the comments, and the manuscript needs to be carefully checked in terms of writing (for example: line 219 has a dot added after the parentheses and should be deleted) and the sentences need to be strengthened in terms of academic writing.

Figure Files:

Data Requirements:

Reproducibility:

References

---

## [Editor Report · Decision Letter 2]

21 Jul 2023

Dear Mrs strang,

We are pleased to inform you that your manuscript 'Gestational toxoplasmosis treatment changes the child's prognosis: a cohort study in southern Brazil' has been provisionally accepted for publication in PLOS Neglected Tropical Diseases.

Best regards,

Hamed Kalani

Academic Editor

Ricardo Fujiwara

Section Editor

---

## [Editor Report · Acceptance letter]

14 Sep 2023

Dear Mrs Gomes Ferrari Strang,

We are delighted to inform you that your manuscript, "Gestational toxoplasmosis treatment changes the child's prognosis: a cohort study in southern Brazil," has been formally accepted for publication in PLOS Neglected Tropical Diseases.

Best regards,

Shaden Kamhawi

co-Editor-in-Chief

Paul Brindley

co-Editor-in-Chief
